# Evaluating the Performance of Proposed Switched Beam Antenna Systems in Dynamic V2V Communication Networks

**DOI:** 10.3390/s23156782

**Published:** 2023-07-29

**Authors:** Tahir H. Ahmed, Jun Jiat Tiang, Azwan Mahmud, Chung Gwo-Chin, Dinh-Thuan Do

**Affiliations:** 1Centre of Wireless Technology, Multimedia University, Cyberjaya 63000, Malaysia; 1211405181@student.mmu.edu.my (T.H.A.); azwan.mahmud@mmu.edu.my (A.M.); gcchung@mmu.edu.my (C.G.-C.); 2Department of Computer Science and Information Engineering, College of Information and Electrical Engineering, Asia University, Taichung 41354, Taiwan

**Keywords:** V2V, vehicular safety, intelligent transportation system, 5G

## Abstract

This paper develops a novel approach for reliable vehicle-to-vehicle (V2V) communication in various environments. A switched beam antenna is deployed at the transmitting and receiving points, with a beam management system that concentrates the power in each beam using a low-computation algorithm and a potential mathematical model. The algorithm is designed to be flexible for various environments faced by vehicles. Additionally, an anti-failure system is proposed in case the intelligent transportation system (ITS) system fails to retrieve real-time Packet Delivery Ratio (PDR) values related to traffic density. Performance metrics include the time to collision in seconds, the bit error rate (BER), the packet error rate (PER), the average throughput (Mbps), the beam selection probability, and computational complexity factors. The proposed system is compared with traditional systems. Extensive experiments, simulations, and comparisons show that the proposed approach is excellent and reliable for vehicular systems. The proposed study demonstrates an average throughput of 1.7 Mbps, surpassing conventional methods’ typical throughput of 1.35 Mbps. Moreover, the bit error rate (BER) of the proposed study is reduced by a factor of 0.1. Additionally, the proposed framework achieves a beam power efficiency of touching to 100% at computational factor of 34. These metrics indicate that the proposed method is both efficient and sufficiently robust.

## 1. Introduction

Intelligent systems are tasked with managing transportation due to their significant impact on daily life, with the goal of full automation through machine and not human involvement. To enhance safety, priority should be given to the implementation and improvement of vehicle-to-vehicle (V2V) and vehicle-to-infrastructure (V2I) technologies [1]. The field of intelligent transportation systems (ITSs) is constantly evolving and V2V communications are being developed as part of them, similar to “connected cars”, which integrate machine-to-machine (M2M) communications and the Internet of Things (IoT) [2]. This information can be contextualized in the context of smart city initiatives [3].

Recently, the research on V2V technology has focused on creating a standardized commercial communication model due to the absence of a universal standard [4]. The four key areas of emphasis are implementation, performance evaluation, new protocol design, and integration. Implementation is necessary for practicality, performance evaluation establishes optimal KPIs, new protocol design standardizes data packet frame, and integration leads to a fully automated ITS solution [5]. V2V networks offer a comprehensive view of road activity through ITS integration, improving upon current solutions like parking and blind spot detection. However, vehicles may move at differing speeds, causing communication problems, and rapid network changes can occur in V2V networks [6].

From an industrial perspective, the main issue in V2V communication is the efficiency of omnidirectional antennas. The study in [7] was conducted using switched beam antennas at the transmitting end and omnidirectional antennas at the receiving end, resulting in improved Received Signal Strength Indicator (RSSI) and probability values for the V2V link, but with power and resource waste affecting the overall channel traffic structure. Algorithms and beam management play a crucial role in enhancing efficiency and improving results. In the above research, both the transmitting and receiving ends were equipped with switched beam antennas for tracking capabilities.

Before considering the experimentation and evaluation process, an initial pre-requisite analysis is discussed. The switched beam antenna consists of 20 or more directional antennas connected or switched through a series of Ethernet commands [8]. These components can be used to generate various predefined beam patterns that cover all directions around the antenna. The sharp azimuthal beams can provide a 360 degree coverage with little variation in maximum power [9]. The switched beam antennas have the ability to cover 360 degrees and provide a higher azimuthal beam coverage compared to Omni antennas [10]. They offer a superior range, coverage, capacity, link reliability, and spectrum efficiency for communication and electric vehicle (EV) applications. For V2V communications, link reliability could be significantly involved. Antenna beam forming and steering are increasingly used in wireless communication systems, particularly in 5G. This technology is advancing to meet the constant demand for faster data rates and higher mobile equipment densities. Beam shaping allows a system of multiple antennas to vary the direction of the overall beam, improving performance by allowing individual users to receive a specific beam and reducing interference. The beam power and computational complexity are significant issues that must be addressed.

For V2V communications, link dependability is the central area of concern. The research on physical antenna constraints can be traced back to Chu’s work published in 1948 [11]. Chu determined the minimal feasible antenna quality factor (denoted by Q), the maximum gain, and the maximum possible ratio of gain to Q for a linearly polarized omnidirectional antenna using the spherical wave function expansion outside the smallest sphere surrounding the antenna. This restriction was also carried forward and, to some extent, is more practical for connected cars.

Furthermore, beam tracking algorithms have numerous different implementations in the literature. Focusing on the core concept of V2V communication, V2V dedicated short-range communication (DSRC) calls for a quick and reliable communication link. From a recent study in [12], RSSI-based beam tracking mechanisms provide the advantages of a low connection loss and straightforward operations. Beam tracking offers a seamless radio link connection for V2V DSRC communication between vehicles. If the beam is appropriately connected for the vehicle’s position on the road, the vehicle can move from one position to another, resulting in an improved signal quality. An enhanced road safety is achieved through a high radio link reliability, a low connection loss, and a smooth connectivity. Vehicles within a communication zone can receive warning messages, preparing them for any dangerous road situations, since the vehicle is constantly connected to the V2V network. Depending on the underlying technology used, V2X communication technology can be divided into two categories:WLAN based.Cellular based

On the other hand, a major flaw observed in this analysis is that in high-speed situations, if the computational complexity increases, significant handover anomalies could occur, resulting in fatal conditions [13].

On the other hand, research has shown that omnidirectional antennas are considered a good choice in vehicles [14]. However, as research progressed, it was found that directional antennas are more powerful than omnidirectional antennas [15]. Despite their greater potency, limitations in their range were observed in mobility models. To overcome this, switched beam antenna systems were proposed, which consist of 20 or more directional antennae that are connected for efficient operation. These systems were mainly used for transmitting vehicles in the V2V use case, with the transmission end powered by omnidirectional antennas [16]. The major drawbacks to current research and approaches are:Each omnidirectional beam radiates power in 360 degrees. Omnidirectional antennas do radiate in 360 degrees, but with a low power quality. This is a concern for highway prompt traffic, where everything should be perfect to avoid fatalities. Additionally, loss of power is observed in the non-overlap beam. This may not be significant in the case of two vehicles, but it will certainly increase the computational complexity and result in powerlessness in a dense or fast vehicular environment.Secondly, an innovative algorithm with low computation between directional antennae in switched beam systems needs to be discussed or proposed.Testing was conducted only in limited environments, whereas vehicular application faces many rigorous environments.

To address the aforementioned issues in this research, an intelligent power management strategy is proposed that utilizes high-quality, focused beams of power. The waste of unnecessary power is reduced by using switched beam antennas at both the receiving and transmitting ends. In addition, an intelligent algorithm with low computational complexity is proposed to switch between beams at both the receiving and transmitting ends simultaneously. The effectiveness of the intelligent algorithms is demonstrated through testing in various environments, including urban, suburban, and highway environments.

Moreover, an anti-failure system has been proposed by incorporating both real-time PDR and non-real-time PDR values. In the event of anomalies in the real-time PDR value, the non-real-time PDR will activate by taking into account the road’s asphalt value according to IEEE protocols and by ensuring the vehicle’s speed falls within a safe range as per the standards for different roads. A comparison between previous research and the research conducted in this paper can be seen in Figure 1.

The computational complexity refers to various aspects that will be discussed in detail later in this paper. In order to address the aforementioned issues, the primary focus will be on addressing heterogeneous complexities. The following contributions will be implemented in this paper:This research focuses on developing a comprehensive switched beam antenna system that addresses the limitations of existing omnidirectional and directional antennas. By utilizing switched beam antennas at both the transmitting and receiving ends, the proposed system optimizes the transmission of power and reduces waste. This enhancement leads to an improved channel traffic structure, an increased transmission efficiency, and an enhanced overall performance of the V2V communication system.This research introduces an innovative algorithm with low computational complexity for beam switching in V2V communication. The algorithm enables simultaneous switching of beams at both ends, utilizing real-time and non-real-time Packet Delivery Rate (PDR) values. By incorporating tracking capability and power switching, the algorithm ensures seamless and efficient beam switching, resulting in improved signal quality, reduced connection loss, and enhanced reliability of the V2V communication link.The proposed research conducts extensive testing and evaluation of the algorithm in various environments, including urban, suburban, and highway scenarios. By conducting experiments in diverse settings, we aim to validate the effectiveness and robustness of the proposed algorithm under different conditions. This comprehensive evaluation helps to establish the practicality and applicability of the algorithm in real-world V2V communication systems.This research proposes a holistic and intelligent approach to V2V systems by integrating the algorithmic-powered solution with an anti-failure system. The anti-failure system incorporates real-time and non-real-time PDR values, considering road conditions and vehicle speed, to ensure reliable connectivity and enhance road safety. By providing warning messages to vehicles within the communication zone, the proposed approach enables proactive measures to be taken in potentially dangerous situations, further improving overall road safety.

Furthermore, the structure of research paper consists of related work in Section 2. Section 3 contains the proposed switched beam steering mathematical model.

Section 4 contains the results and discussion. Furthermore, Section 5 contains the conclusion.

## 2. Related Work

Beamforming is the process of creating an energy beam using phased arrays, which involves adjusting the antenna spacing and signal phase of each element in the array. This allows for control over the size and direction of the signal beam from multiple antennas [17]. Beamforming involves interference and pattern construction, and beam steering is an advanced technique that dynamically modifies the beam pattern by altering the signal phase in real time without changing the hardware [18]. Beam steering is commonly used in 5G and Wi-Fi to direct the radiation or reception beam to a specific station, maximizing gain for that station and minimizing interference with others. These concepts are fundamental for vehicle communications [19].

The increasing demand for voice-over-internet protocols, on-demand bandwidth, and multimedia applications, especially in connected cars, necessitates small, portable antennas with high-speed and low-latency wireless connections. This study evaluates various 5G beamforming techniques, such as analog, digital, hybrid, switching, and adaptive systems, as well as their types, working algorithms, antenna design, gain, and substrate characteristics [20]. These techniques generate different beam widths for specified directions.

As wireless and telecommunication protocols for vehicle-to-vehicle (V2V) communication advance, artificial intelligence, machine learning, and deep learning are playing a significant role. A study [21] proposes an optimal Minimum Mean Squared Error (MMSE) receiver-based heuristic solution structure for downlink beamforming. This study focuses on power allocation and virtual uplink beamforming (VUB) using the Beamforming Prediction Network (BPNet). The proposed method achieves efficient performance in high-speed environments for intelligent transportation systems (ITSs) [22].

Another study [23] investigates the benefits and drawbacks of using beamforming for beaconing in V2V communication. Simulation results show that while beamforming increases collisions, it also enables reaching a higher number of vehicles in the network, leading to more reliable communications. However, the study does not consider the effect of increased power usage on channel loading.

As part of the related work, the utilization of physical layer security techniques, such as friendly jamming (FJ), has been explored to enhance the confidentiality of data uploading in mobile edge computing (MEC) services. FJ has shown promising results in degrading attackers’ wiretap channel capacities and effectively protecting users from potential threats [24]. In this paper, the authors propose a comprehensive security scheme that incorporates FJ as a physical layer security mechanism. This scheme involves the coordinated use of FJ from both the base station (BS) and nearby mobile users to cover the upload transmission, thereby ensuring enhanced confidentiality in MEC services.

One drawback of this FJ-based physical layer security mechanism is the potential increase in resource utilization and communication overhead due to the additional coordination and signaling required for implementing FJ.

The emerging concept of the Internet of Vehicles (IoV) in the 5G era has gained significant attention. One study [25] focused on multicast beamforming and multimedia content caching in IoV-based vehicle edge networks. The authors addressed the optimization of clustering, caching, and multicasting with V2V assistance to meet Quality of Service (QoS) requirements. However, the study did not test the power cost in high-speed environments.

Enhancing V2V communication effectiveness is explored in [26] through beamforming and scalable transmission. Beamforming increases the bandwidth by establishing clear and reliable beam connections between vehicles. Scalable transmission reduces data volumes by transmitting only relevant information. However, the study acknowledges potential problems in high-speed scenarios if power management and intelligent algorithms are not considered.

Localization is a critical challenge in V2V communication, and the relationship between beamforming and beam alignment is examined in study [19]. This study incorporates realistic vehicle dynamics and highlights the limitations of the Kalman filter for non-linear parameters. Hardware limitations on antenna elements and phase shifters are also considered.

Directional antenna designs and beam steering play a crucial role in providing reliable and secure V2V communication, as studied in [27]. This study also addresses the Doppler effect and introduces an NLOS mode of operation for specific services.

In another study [28], a customized beamwidth optimization problem is formulated to prevent beam misalignment in V2V communication. The authors propose a Monte Carlo-based beamwidth optimization (MCBO) method to address the issue statistically.

While the aforementioned research studies provide valuable insights, there is a need for practical simulations and algorithm testing in various environments, including highways and suburban and urban areas. It remains unclear how V2V communication would perform in different scenarios, and different conditions can affect the reliability of algorithms and mathematical models. Furthermore, although beam steering and beamforming techniques have been extensively studied, their implementation in V2V scenarios requires further exploration. Therefore, this research aims to design a new algorithm and mathematical model for beam steering with a tracking ability, which will be tested in three environments (highways and urban and suburban areas) to evaluate the reliability and computational complexity of the algorithm for V2V links.

## 3. Proposed Mathematical Model of a Switched Beam Antenna with Beam Steering Abilities

Focusing on optimizing the half power bandwidth (HPBW) and the first null bandwidth can improve the power concentration for the beam-to-beam management and V2V links. The concept of the HPBW and the FPBW can be illustrated in the accompanying Figure 2.

The FNBW plays a role in the antenna radiation pattern, which can be considered as the angular breadth between the first side lobes or nulls. The resolution capacity of the antenna, which defines the system’s ability to distinguish between two adjacent targets, is determined by the beam width. In the scenario of V2V communication, two targets are represented by two vehicles, and differentiation between them is important to prevent collisions. The factors that affect the beam width include:The shape of radiation patterns.The antenna dimensions.The wavelength.

Furthermore, for power calculations of a fine beam, the HPBW and FNBW are both considered for concentrated beam power analysis. For V2V links, the SNR ratio should be maximized, followed by a high RSSI value. In this paper, a five-switched beam system with optimum and combined values of HPBW and FNBW is observed to enhance SNR and RSSI values. A comparative analysis will be conducted later on the amount of power transmission. The antenna gain function depends on beam directions, as reported in reference [29].

R(safe) indicates the distance the driver has left to prepare before entering a dangerous situation on the road. Additionally, the equation links the suggested system to the time to collision or safety time (Tsafe). Therefore, the Effective Communication Range (RECR) of the antenna technology is crucial for the V2V communication link, and both (Rsafe) and (Tsafe) are essential for analysis.
(1)$$R={\left(Z-0.355\right)}^{2}/99.8$$
(2)$$Tsafe=(R\left(safe\right)-{\left(R-D\left(ped\right)\right)}^{2}/Z$$

D(ped), which refers to the pedestrian distance from a specific vehicle, was added. This is essential to address after the Uber fatality case [30].
(3)$$Tsafe=(R\left(safe\right)-{\left(R-D\left(ped\right)\right)}^{2}/Z$$

The relationship between the transmitting and receiving end embedded by the beam switching antenna can be shown as:(4)a=b(x1+x2+x3)+n
where *a* represents the receiving antenna and *b* represents the wireless channel with different propagations. The power estimation of a specific beam is a crucial step and will be carried out by considering the transmitted data with respect to the channel propagation.
(5)$$Power=\left(X^{2}*B^{2}+N^{2}\right)*HPBW+\left(X^{2}*B^{2}+N^{2}\right)*FNBW$$

Similarly, the received power with respect to beam switching is as follows:(6)Power=Transmitted power∗propagation∗noise

Beam communication is depicted in Figure 3.

The channel coefficient can be determined by Friis transmission [31]. The angle of transmission with respect to the HPBW and FNBW is shown in Equations (7) and (8).
(7)$$\begin{array}{cc} Power(Angle)\; & =g\left(i\right)\cdot 10^{0.1}(G\left(max\right)-\\ \; & 12[\theta (Power)-\theta (Deviation)/B.W] \end{array}$$
(8)$$\begin{array}{cc} Power(Angle)\; & =g\left(i\right)\cdot 10^{0.23}(G\left(max\right)-\\ \; & 18[\theta (Power)-\theta (Deviation)/B.W] \end{array}$$

Similarly, the estimation of the probability of power with respect to the vehicle is as follows:(9)P=P(v)/Difference of angel<Threshold value
For an appropriate beam switching antenna configuration, a Fourier analysis will be used:(10)$$J\left(i\right)=\left[1/\sqrt{N}\right]\cdot e^{-j2\pi /Nf(i-Nf+1/2)}$$
(11)$$Fine\;beam=J\left(i\right)=\left[1/\sqrt{N}\right]\sum e^{jw.p}$$

The Doppler frequency F(doppler) depends on the relative velocity *V* between the transmitter Tx and receiver Rx, the speed of light Co, and the carrier frequency Fo for fast-moving vehicles with all developed angles with respect to the power of the beam and fine beams FB.
(12)$$\begin{array}{cc} F(doppler)\; & =[(V/Co)*Fo*Cos(\alpha +\beta +\\ \; & +\gamma )*FB-P(beamswitch)] \end{array}$$
(13)$$\begin{array}{cc} F(effect\;in\;particular\;vehicle)\; & =[F(doppler)\\ \; & -F(Fine\;beam)]/(particular\;angel) \end{array}$$
(14)Coherence time=8.78/16.23∗pi∗(f(total)−f(effected)
(15)F(doppler)=(V/Co)∗Fo∗Cos(α+β+γ)∗Finebeam

The mathematical model mentioned above initially calculates the optimum concentration of the beam-to-beam connection from one vehicle to another, taking into account the speed and propagated noise in a specific environment. Additionally, after the beam-to-beam power calculation, it analyzes a specific angle with respect to threshold values for a safe wireless connection. Furthermore, the establishment of a fine connected beam is observed after the power calculation, taking into account the Doppler effect, especially for high-speed scenarios. Mathematical modeling plays a crucial role in making a breakthrough in developing an intelligent testbed for real-time results investigation.

After successfully modeling a secure V2V power loss model, a testbed for the algorithm is established. The model is transformed into code, which combines C++ and Python, with the aim to:Be adaptable to various environments.Contain beam switching with respect to threshold power according to the mathematical model.Be computationally friendly with a low complexity factor.

For environmental detectability, two packet rate delivery values are taken into account: a real-time scenario-based PDR and a PDR based on IEEE protocols considering the road’s asphalt value. The first value indicates the real-time traffic density of the situation and determines whether it is an urban, suburban, or highway scenario. In an urban scenario, beam switching between two vehicles would be low powered to reduce power losses and address network complexity and channel congestion. Additionally, the second PDR value is considered if the first value fails due to issues in the arrival system. If fundamental time data are not collected due to issues with high-speed traversing, the algorithm will switch from real time to the road asphalt value (which is different for different environments) to adjust to the appropriate levels of power beam, avoiding fatalities. The workflow for the environment settings can be observed in Figure 4. The code flow can be observed in Algorithms 1 and 2.

The effective communication range (ECR) or R(safe) is established based on two parameters that are fed into the algorithm: PDR based on the road asphalt value according to the IEEE 802.11p standard and PDR based on the absolute traffic density. This ensures that the connection reliability is guaranteed when a vehicle moves through a communication area. The PDR value of 0.9 and the vehicle speed *Z* are related to the safety range R(safe). Similarly, the PDR value in this research is set at 0.88 for suburban roads and 0.94 for highways. The exact values for real-time PDR are also set. In the case of disconnection, two PDR values are considered. If there is any disruption and the real-time PDR is miscalculated, the actual road situation is sensed, the PDR based on asphalt value is configured, and surrounding vehicles are shifted to manual sensors to avoid fatalities and establish a controlled environment. The algorithmic structure is described in Algorithms 1 and 2.
**Algorithm 1** Beam switching with PDR setting via a selection window**Require:** **Input:** *choose_resource_selection(self, reselection_window)***Ensure:** **Output:** *reselection_window += 1*1:V2V_P2:PDR_threshold3:$min\_PDR=\frac{\mathrm{len}(selection\_window)}{4.67}$4:PDR←counter5:**while** len(PDR)<min_PDR **do**6:    PDR←counter+17:    **for** beam_selectioninrange(len(selection_window)) **do**8:        **if** len(PDR)=0.88 **then**9:           V2V_P←P(least_urban)            ▹ Fine Power Beam10:        **else if** 0.92≤len(PDR)≤0.99 **then**11:           V2V_P←P(moderate_sub_urban)12:        **else**13:           counter+1                    ▹ Highway settings14:        **end if**15:    **end for**16:**end while**17:sorted_R_PDR←sorted(PDR.items(),key=λx:x[1])18:min_len←min(min_PDR,len(PDR))19:**for** kinsorted_R_PDR **do**20:    sB.append(k)21:    **if** len(sB)≥min_len **then**22:        **break**23:        random.choice(sB)24:    **end if**                    ▹ Real Time PDR Execution25:**end for**

**Algorithm 2** Real-time value upgradation and switching via RSSI values for a V2V link
**Ensure:** def_step(self,selection_window): param_selection_window: action=0; **if** selection_counter!=0 **then**     Use previous selected packet index and reduce reselection counter.     Action=V2V_linkeage;     self.reselection_counter−=1; **else if** random()<self.prob_resource_keep; **then**     action!=V2V_linkeage **else if** $action=self.choose_{n}ew_{r}esource\left(selection\_window\right)$ **then**     self.prev_action=continue **else**     The V2V link is inadequate due to a lack of resources. **end if** return_action;


This research has algorithm settings as its central part. Integration of several tools has been carried out, including the Python Development Kit and NS3. Additionally, three significant developer stacks have been created: front-end, middle stack, and back-end.
The front end contains Doppler range selection according to the proposed mathematical model for various environments.The middle stack contains beam switching based on power concentrations to maintain the V2V link quality to meet the standards.A central management algorithm system is created to control the overall communication between different components.

The in-depth structure can be illustrated in Figure 5. The parameter settings of the conducted research are shown in Table 1.

Synchronization errors in V2V communication systems can arise from several technical factors, leading to misalignment between the transmitting and receiving ends. These errors can have significant implications on the overall system performance:

Propagation Delays: Propagation delays caused by the finite speed of electromagnetic waves can introduce synchronization errors. The variation in the distances between transmitting and receiving vehicles can result in different signal arrival times, leading to misalignment of beamforming and beam tracking operations.

Clock Drift: Clock drift in the transmitting and receiving devices can introduce timing discrepancies, causing synchronization errors. Inaccurate clocks or drift over time can lead to misalignment of the beamforming intervals, resulting in reduced beamforming efficiency.

Channel Estimation Inaccuracies: Inaccurate estimation of channel characteristics, such as channel gains and phases, can contribute to synchronization errors. Mismatches in channel estimation can lead to misalignment between the intended beamforming direction and the actual received signals.

Hardware Limitations: Limitations in the hardware components, such as processing delays or latency, can also contribute to synchronization errors. Delays introduced during signal processing stages can affect the alignment between the Tx and Rx ends, impacting the accuracy of beamforming operations.

These synchronization errors can degrade the performance of V2V communication systems, leading to suboptimal beamforming, a reduced signal quality, and a decreased communication efficiency. Mitigation techniques such as time synchronization protocols, advanced channel estimation algorithms, and hardware improvements are essential to address and minimize these synchronization errors, ensuring accurate and precise synchronization between the transmitting and receiving ends.

To address synchronization errors in V2V communication systems, several techniques can be employed. Firstly, implementing robust time synchronization protocols, such as the Precision Time Protocol (PTP) or Network Time Protocol (NTP), ensures accurate time synchronization between transmitting and receiving devices. This helps minimize propagation delays and clock drift, improving synchronization accuracy. Secondly, advanced channel estimation algorithms can be utilized to accurately estimate channel characteristics, including gains and phases. By compensating for channel estimation errors, misalignment between the intended beamforming direction and the received signals can be reduced, enhancing synchronization.

Additionally, hardware improvements play a crucial role in mitigating synchronization errors. Upgrading hardware components involved in the V2V communication system, such as high-precision clocks and low-latency signal processing units, minimizes synchronization errors arising from hardware limitations. Lower processing delays and an improved precision contribute to more accurate synchronization. Furthermore, adaptive synchronization mechanisms can be employed, which dynamically adjust the synchronization parameters based on real-time feedback and system conditions. This adaptive approach helps mitigate synchronization errors caused by varying propagation delays and clock drift, maintaining accurate synchronization under changing environments. By implementing these techniques, synchronization errors in V2V communication systems can be effectively addressed, leading to improved beamforming performance, enhanced signal quality, and increased communication efficiency. However, it should be noted that the analysis of synchronization errors in real-time scenarios falls outside the scope of this study, which primarily focuses on simulative analysis.

## 4. Results and Discussions

The experimental setup entails a simulated environment where a transmitting vehicle and a receiving vehicle are examined, both equipped with reflectors of varying half power bandwidth (HPBW) values. The antenna angle for steering and tracking is configured at 14 distinct angles: 0, 30, 45, 60, 75, 90, 120, 150, 180, 210, 235, 270, 310, and 360 degrees. Notably, these angles are not set at four intervals, as the receiving end employs a switched beam antenna instead of an omnidirectional antenna. The algorithm is executed at both the transmitting and receiving ends. In accordance with the DSRC standard, the transmitting beam steers every 100 ms, and the receiving beam follows the same rate, facilitating improved connectivity between moving vehicles.

The Packet Delivery Rate (PDR) depends on two parameters: the current situation’s density (congested or not), received from the base station, and the road’s asphalt value, as specified by the DSRC for different environments. Optimizing the PDR is crucial for reducing the power and computational complexity while enhancing the power connection over long distances or in power-constrained scenarios. In the experimental setup, a lane-changing scenario is observed, where a beam-to-beam connection is established using a message size of 1 KB. The utilization of beam angle slicing assists in establishing low-power or long-distance connections, which constitutes the main focus of this research.

The first performance metric is the Time to Collision, relative to the vehicle speed, as depicted in Figure 6. By optimizing the Full HPBW and First Null Bandwidth (FNBW), denoted as T(safe), in comparison to conventional approaches, notable improvements are achieved, primarily attributed to the enhanced operation of the dual-side switched beam antenna. The algorithm, designed to monitor various crucial parameters and adapt to the situation, demonstrates that at a vehicle speed of 120 km/h, the promptness of the warning message increases by 4 s. This signifies a more efficient delivery of warning messages compared to the previous methodology, significantly reducing the computational power required by the system.

The quality of V2V communication links directly affects the bit error rate (BER). In comparison to conventional systems [7,32,33], the proposed system demonstrates a more efficient BER. To evaluate the system’s performance, a high-speed lane-changing scenario is chosen, where both the On-Board Units (OBU) function as both transmitters and receivers, employing a switched beam system. The distance between the two vehicles is set at 500 m, with an average speed of 100 km/h. The transmission packets consist of 1 KB, containing warning messages as the core information.

Unreliable V2V links with high bit error rates (BER) compromise traffic safety, as incorrect warning messages may be received. Throughput, another critical Quality of Service (QoS) metric, can be estimated using the approximation method, as illustrated in Figure 7.

The packet error rate (PER) is a vital metric for evaluating the efficiency of the system. It is defined as the ratio of the number of error packets, after Forward Error Correction (FEC), to the total number of received packets. A packet refers to a data unit transmitted through radio waves and is subjected to FEC. In comparison to previous research, the proposed system exhibited a 10% improvement in the PER, as depicted in Figure 8. This improvement directly enhances the transmission reliability. The notable factor contributing to the PER improvement is the utilization of a low computational power and lightweight algorithmic systems in the system.

The throughput analysis demonstrates that the proposed system achieves faster message delivery compared to the conventional system. Algorithmic design and computational complexity play a pivotal role in this regard. Omnidirectional antenna systems necessitate computationally complex algorithms due to the four-dimensional management of antenna power. Conversely, switched beam systems at the receiving end require fewer power resources as they employ low power slicing, thereby maintaining a low wireless power at all points. The results illustrating these findings are presented in Figure 9.

The beam selection probability depends on the beam switching between different directional antennas. The probability of transitioning from one beam to another in a highly mobile environment is significant. In Figure 10, after algorithm deployment, beam switching is observed. The average beam switching probability is 0.9, due to the reduced complexity, which is considerable and reliable for fast-paced environments.

Furthermore, the computational complexity factor is calculated to observe the effect of the algorithm in various environments. The computational complexity factor is directly related to the traffic density and its speed analysis, taking into consideration the network point load and channel load measurements. According to Figure 11, the computational complexity is significantly decreased in increased traffic. The average computational factor in a highly dense environment is 0.29. In regular practice, the computational factor reaches a value of 0.4 in dense environments, resulting in a decreased performance or the halting of the analysis of intelligent transportation systems.

In analyzing beam efficiency and its power transfer to pinpoint locations, a new algorithm and mathematical model have demonstrated a significant decrease in computational complexity. The beam efficiency at more than 90% has a complexity factor of 15, which is desirable enough for any dense traffic environment. On the other hand, the conventional system was not consistent in improving the complexity factor and tended to average at 0.25. The results are illustrated in Figure 12.

The connected vehicular density is a critical aspect to consider. This is determined by several key parameters, such as the connected links, the density of a 500 m roundabout area as a test scenario, and the difference in related links between the traditional and proposed systems. Additionally, the impact of increased traffic on the performance parameters of virtual servers is evaluated for performance estimation. In Figure 13, it is observed that the proposed system and its algorithmic structure, with support for various environments, are resource friendly.

The effective communication range of DSRC demonstrates the success of safety messages received in long-range communications, and it also takes into account the driver’s control of the vehicle in case of communication failure. Communication was observed after 15 s, 15 min, and 28 min, with the scenario and density of the vehicular system remaining stable and converging to an ideal state. The proposed system and its algorithmic structure were robustly compared to conventional systems, with the average range of the proposed architecture converging at 872 m, while the conventional systems converged at 869 m. Despite having competitive results, the conventional system was surpassed by the proposed novel architectural design, which showed a 3 m improvement in range on average after 28 min of connectivity. In fast-paced vehicular systems, even a small and reliable improvement in range, such as 0.1 m, can make a significant difference. The results can be seen in Figure 14.

Another strategy is to determine the quality of V2V links and their impact on computational resources. The V2V quality factor incorporates two parameters: range effectiveness and connectivity time in a harsh environment (fast lane change is considered the worst condition in this study). The computational complexity factor, as mentioned before, represents the performance range of available virtual servers. A computational complexity factor of 3 indicates that resources are highly utilized. In the case of a high complexity factor, the quality of the proposed system is better by two factors, as shown in Figure 15. This result is due to a low-resource key code and model optimized for GPU machines.

The time complexity of an algorithm refers to the total time required for the algorithm to complete its execution. In simpler terms, it is the amount of time it takes for a piece of code to run. The faster the code executes, the lower its time complexity. The proposed system has demonstrated exceptional results, as illustrated in Figure 16.

Furthermore, in anti-failure vehicular systems, if the real-time PDR fails to fetch data, the PDR is calculated with respect to the asphalt value of the road environment according to standards [34]. In that scenario, the success of the algorithm’s execution is also observed to be part of the proposed system, as shown in Figure 17. The individual vehicular node, the transfer rate in Kbps, and a directional analysis are depicted in Figure 18.

Furthermore, the proposed method can improve the computational issues of connected cars in complex and dense environments, especially in underdeveloped nations where incidents occur in large numbers. Moreover, the beam management strategy, which takes into account the power density, significantly reduces power losses. The potential of using an omnidirectional antenna is also explored to assess the challenges faced by switched beam antennas. The proposed algorithm is competitive in various environments and conditions, as demonstrated by different parametric and evaluation metrics. Additionally, the anti-failure system that directly relates to the asphalt value is a crucial aspect of safety.

## 5. Conclusions

In conclusion, this paper presents a novel approach to V2V communication that combines low-computational algorithms with a high efficiency, tailored for diverse environments. This anti-failure strategy significantly enhances vehicular safety by mitigating the risk of accidents. The utilization of switched beam antennas has proven to be highly effective, with a minimal power consumption. Through extensive simulations and experiments, the proposed strategy has demonstrated an exceptional performance, surpassing the limitations of current techniques.

The results indicate that the integration of algorithms with switched beam antennas enhances the efficiency of V2V networks from several aspects. These include a reduction in the packet error rate, an improved beam selection probability, a stable average throughput, a lower bit error rate, and an improved time to collision compared to non-algorithmic or high-computational-resource approaches. Furthermore, the incorporation of mathematical models into the algorithms enhances the beam power efficiency, particularly in switched beam scenarios with limited computational power. This research provides cutting-edge and efficient solutions for the latest vehicular networks, significantly enhancing their performance.

For future work, the integration of distributed ledger technology into 5G can be explored to address privacy concerns in V2V communication, ensuring data integrity and preventing tampering. Additionally, it can provide protection against reverse cyberattacks at the DSRC message exchange nodes, further enhancing the security of the system.

This study primarily focuses on a simulative analysis, which may not fully capture the complexities and dynamics of real-world V2V communication environments. Real-world factors such as varying traffic conditions, interference, and non-ideal channel conditions are not fully accounted for in the simulation.

## Figures and Tables

**Figure 1 sensors-23-06782-f001:**
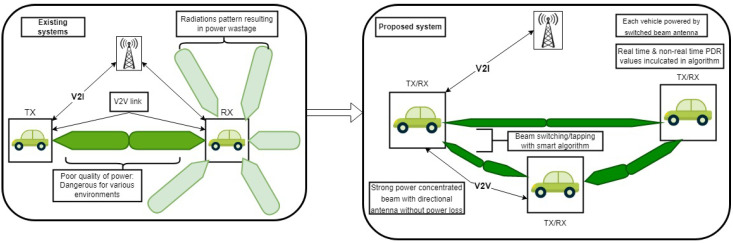
Comparative structural analysis of existing and proposed architectures.

**Figure 2 sensors-23-06782-f002:**
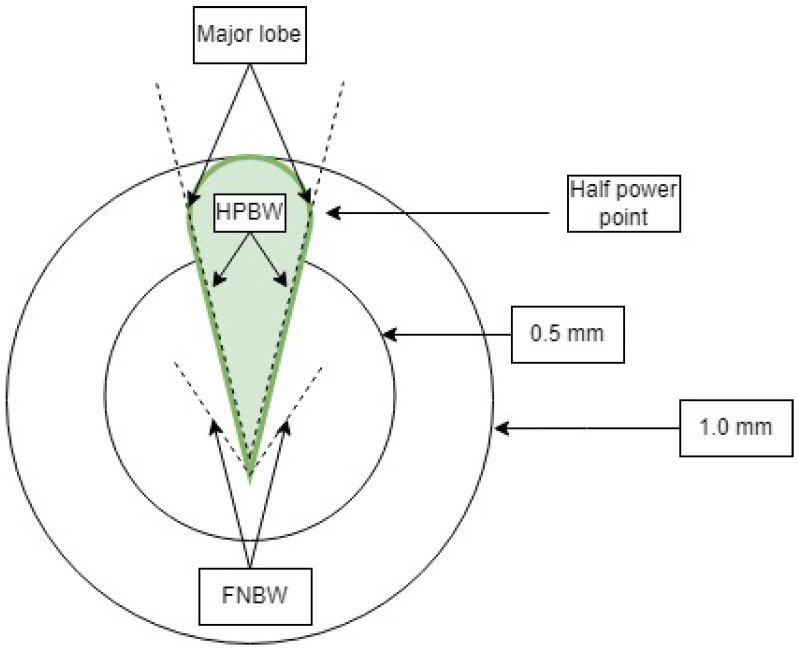
Structural explanation of HPBW and FNBW lobes.

**Figure 3 sensors-23-06782-f003:**
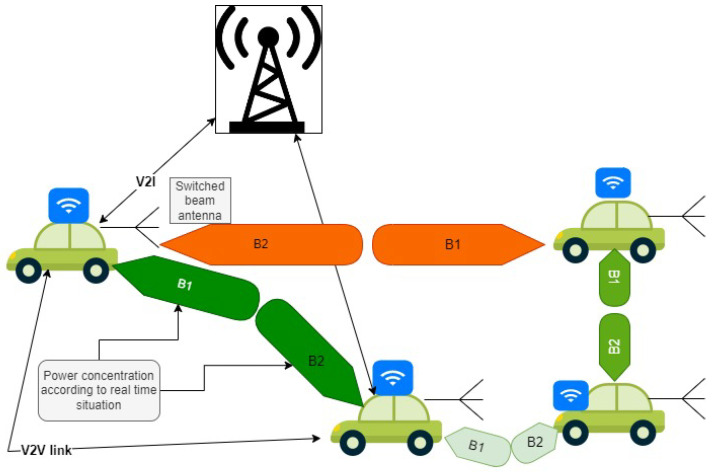
Proposed beam-to-beam power connection.

**Figure 4 sensors-23-06782-f004:**
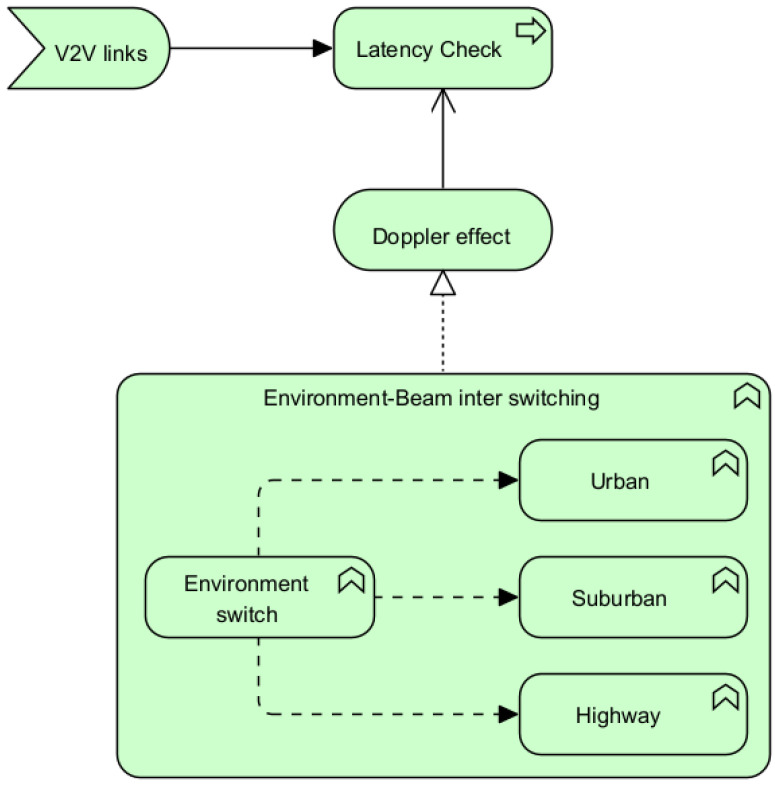
Concept of the proposed algorithm in case of real-time PDR failure.

**Figure 5 sensors-23-06782-f005:**
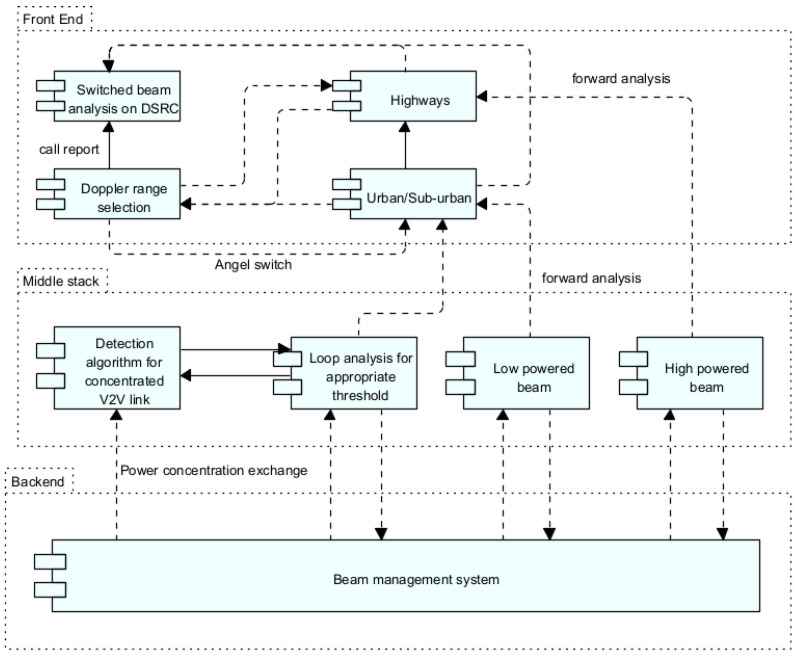
Experimental and working structure of the algorithm.

**Figure 6 sensors-23-06782-f006:**
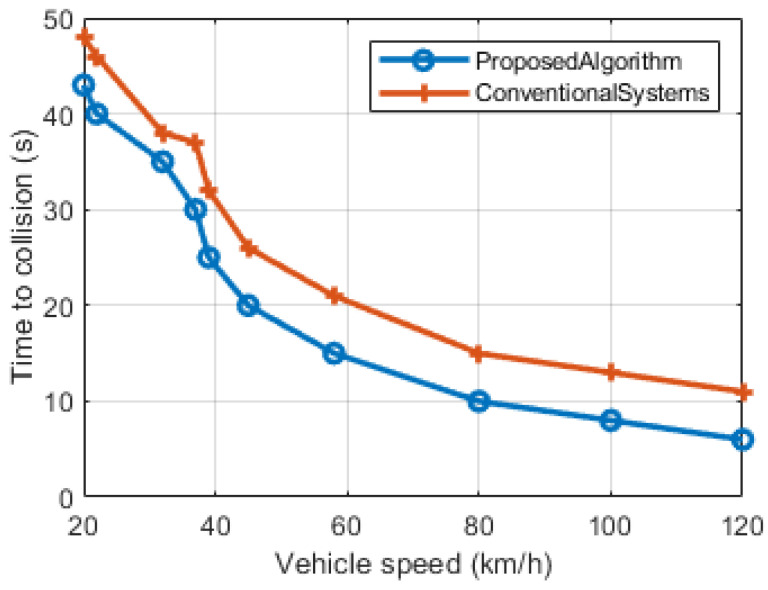
Time to collision in seconds with respect to vehicular speed.

**Figure 7 sensors-23-06782-f007:**
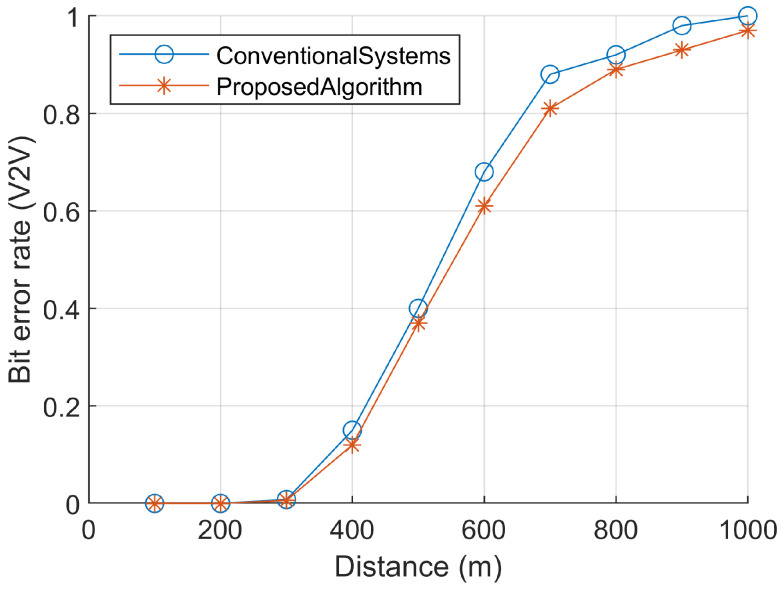
BER calculation with respect to distance in meters.

**Figure 8 sensors-23-06782-f008:**
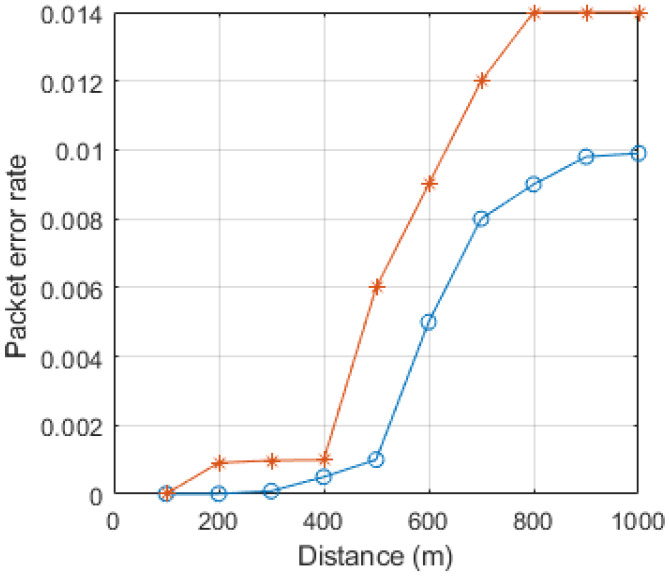
PER calculation with respect to distance in meters.

**Figure 9 sensors-23-06782-f009:**
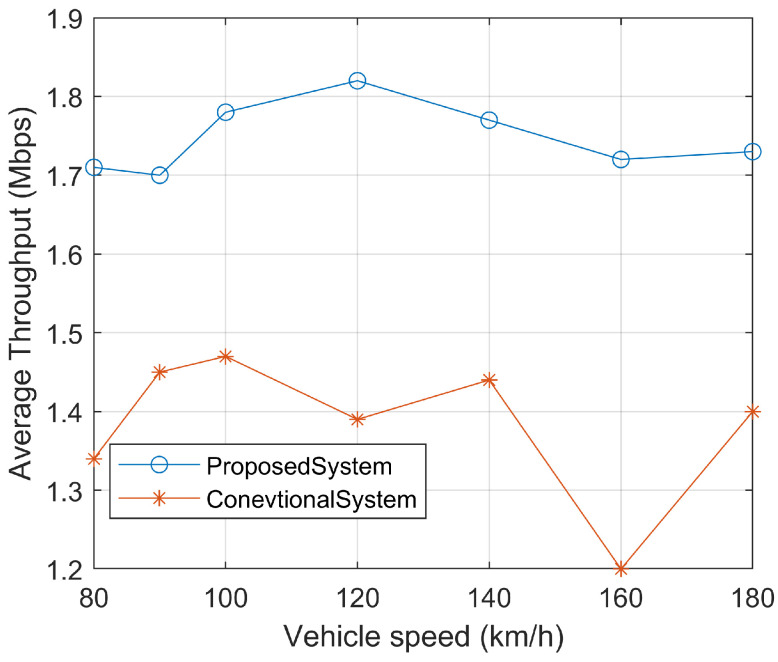
Calculation of average throughput in Mbps.

**Figure 10 sensors-23-06782-f010:**
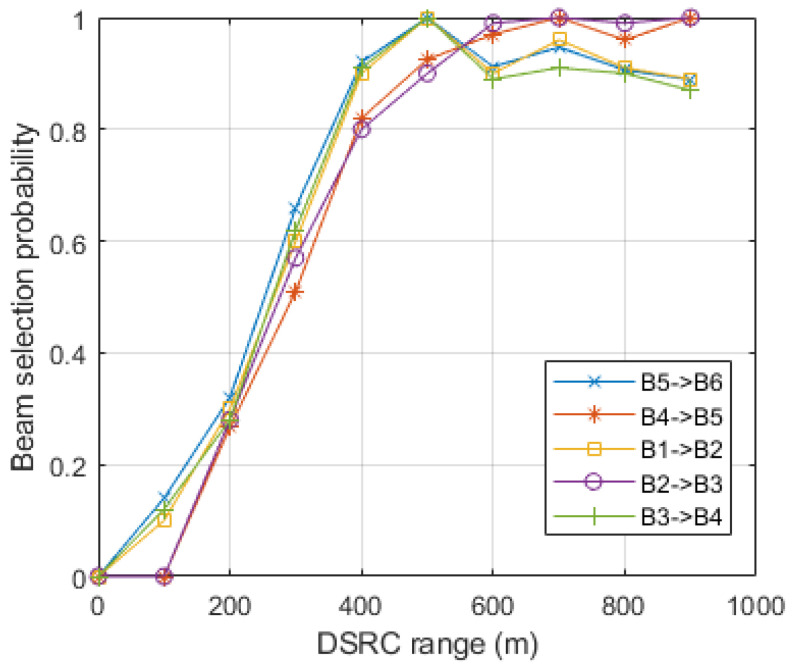
Beam switching and selection probability liaise with DSRC range.

**Figure 11 sensors-23-06782-f011:**
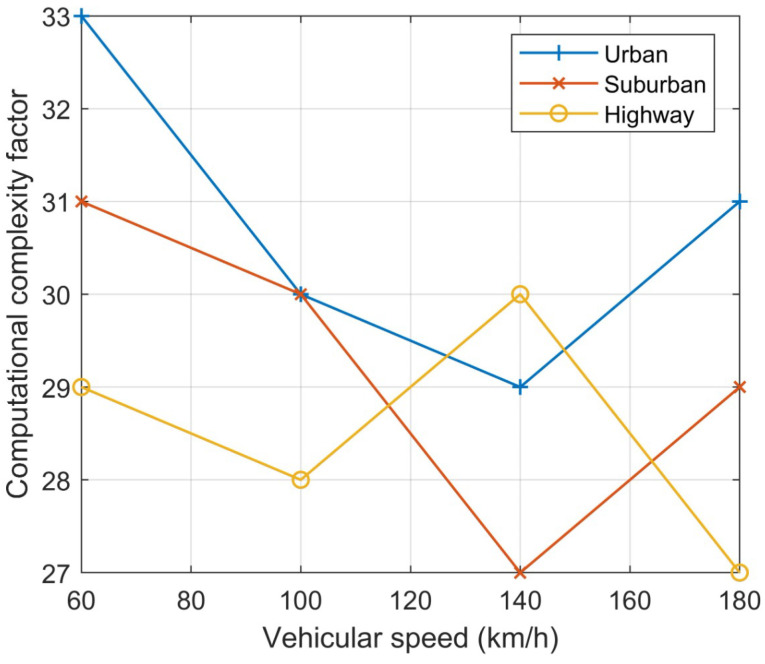
Calculation of computational complexity in various environments.

**Figure 12 sensors-23-06782-f012:**
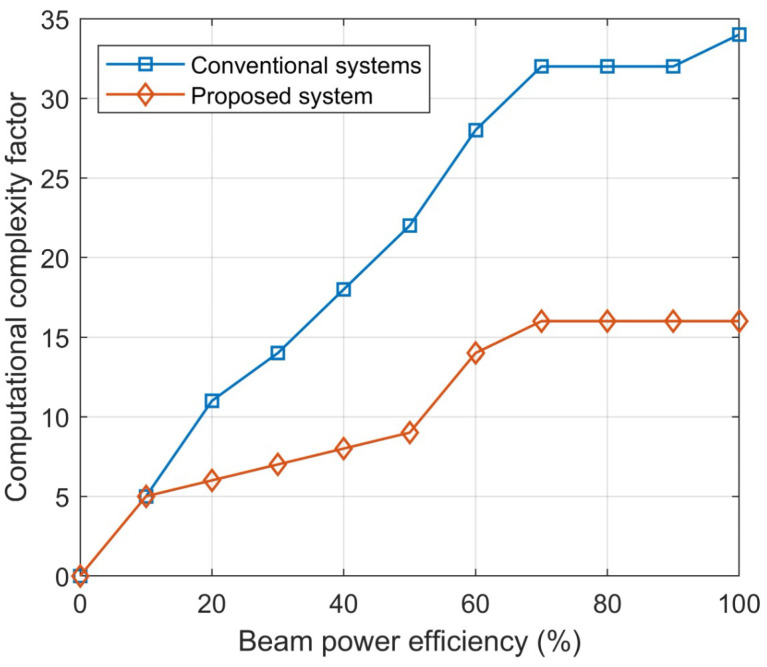
Estimation of the computational complexity factor with respect to beam power efficiency.

**Figure 13 sensors-23-06782-f013:**
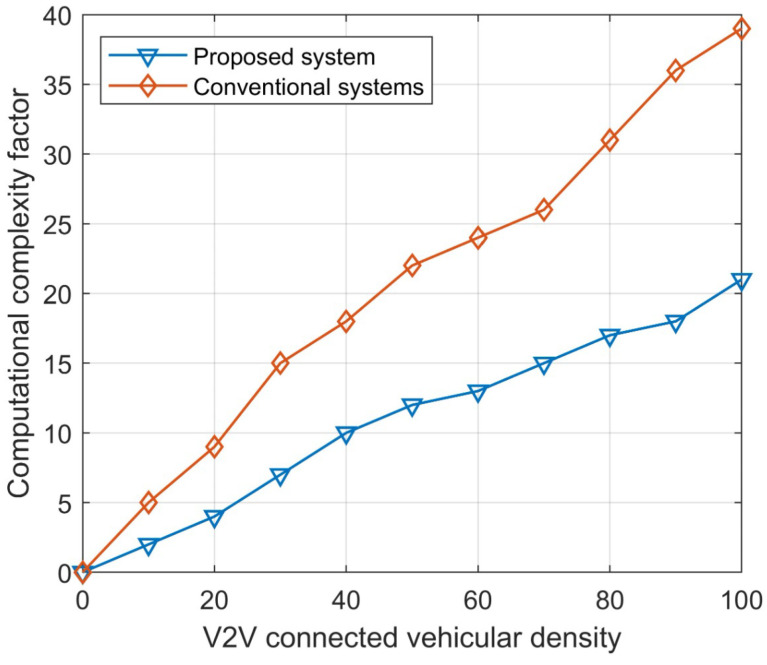
Estimation of the computational complexity factor with respect to vehicular density.

**Figure 14 sensors-23-06782-f014:**
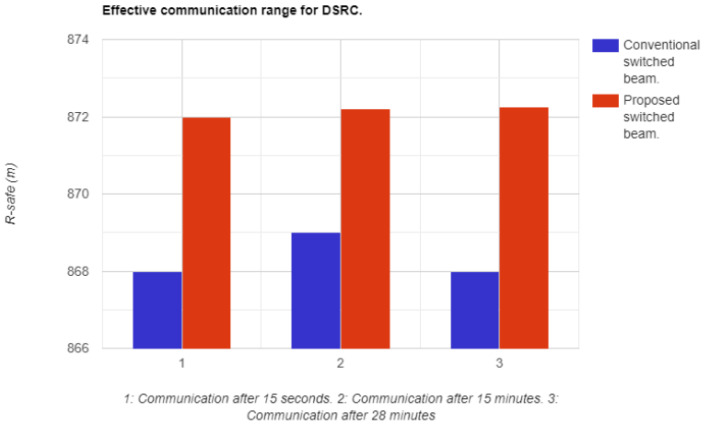
Communication range between vehicles at different time intervals.

**Figure 15 sensors-23-06782-f015:**
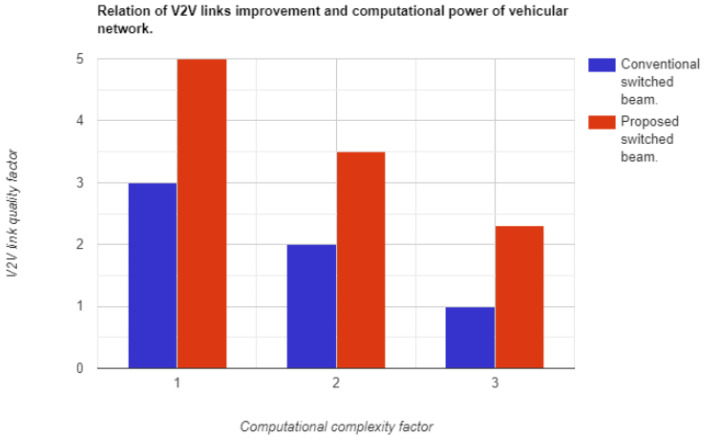
Effect of computational complexity on V2V link quality.

**Figure 16 sensors-23-06782-f016:**
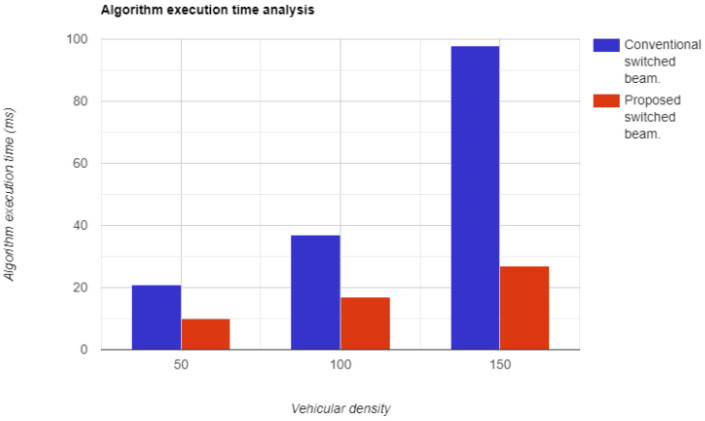
Algorithm execution time with respect to vehicular density.

**Figure 17 sensors-23-06782-f017:**
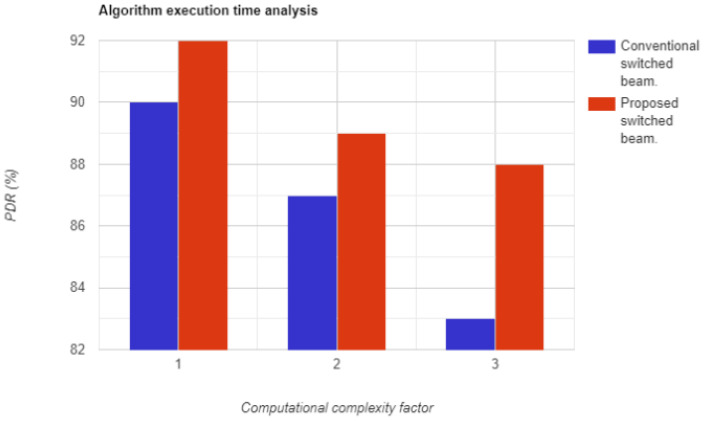
Algorithm execution time in an antifailure algorithmic system.

**Figure 18 sensors-23-06782-f018:**
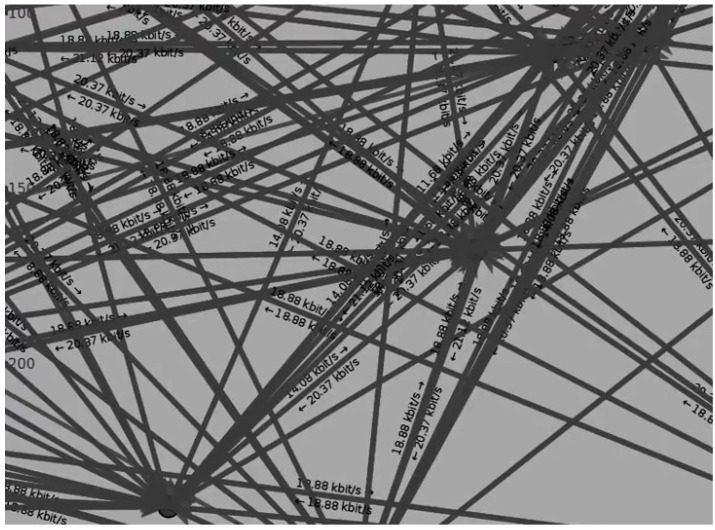
ns3 vehicular node visualization with message communication data rate.

**Table 1 sensors-23-06782-t001:** Parameter settings.

Parameters	Value (s)
Number of Nodes	500
HPBW Antenna Angles	0, 30, 45, 60, 75, 90, 120, 150, 180, 210, 235, 270, 310, 360 degrees
Transmitting Interval	100 ms
Message Size	1 KB
Distance between Vehicles	500 m
Average Speed	100 km/h

## Data Availability

Not applicable.

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
