# Peer review of "Evaluating the Performance of Proposed Switched Beam Antenna Systems in Dynamic V2V Communication Networks"

_sensors, 2023, doi:10.3390/s23156782_

Round 1
Reviewer 1 Report
This paper investigates reliable vehicle-to-vehicle (V2V) communication in various environments and proposes an innovative algorithm with low computational complexity for beam switching in V2V communication. The results indicate that the integration of algorithms with switched-beam antennas enhances the efficiency of V2V networks in several aspects. The reviewer has the following major comments:
1. What is the relationship between Algorithm 1 and Algorithm 2? Is the result of the experiment part of Algorithm 1 or Algorithm 2?
2. In simulation part, there are a few experimental comparison schemes, which need to be compared with other research works.
3. The parameter settings of the experiment should be provided.
4. Some latest studies about V2V are missed in the related work, e.g., Distributed Multi-Agent Reinforcement Learning for Cooperative Edge Caching in Internet of Vehicles, IEEE TWC. Furthermore, the authors are advised to improve the quality of all images, such as Fig. 1, Fig. 3 and so on.
Some typos and grammar mistakes exist in this paper, the authors need to check the whole manuscript carefully.
Author Response
Comments are addressed in PDF file.

Reviewer 2 Report
The authors have presented Evaluating the Performance of Proposed Switched Beam Antenna Systems in Dynamic V2V Communication Networks. The following are observations/suggestions/possible modifications:
1. Abstract should have more statistical information. The performance parameters may be incorporated.
2. Page 2 Line 77: It is stated that a major flaw observed in this analysis… Can you please cite some references or case studies here to show the flaw?
3. Figure 1 could be shown a little larger. The text within the image is not fully visible.
4. Overall, the introduction is satisfactory. The citations are arranged in such a way that flow is maintained. Please ensure the presence of the latest references.
5. Page 4 Line 155: The figure number is missing in the line immediately after reference 18. Please include.
6. Figure 2 could be improved for the text.
7. Page 5 Line 203: Highways, urban and suburban mentioned in the brackets. Avoid writing in the brackets in the entire manuscript other than notations viz. Rsafe, Tsafe, etc..
8. Page 6 Line 218. It has to be Antenna dimensions and Wavelength with W capital letters.
9. Equations need citations if they are not defined by the authors.
10. Figure 3: Text needs improvement
11. Page 7 Line 249: notations C0 and f0 are not correctly shown.
12. Figure 4: fonts need to be reviewed for better visibility.
13. Page 9 Line 281: Please verify Rsafe – it is mentioned as R – Safe.
14. Page 10 Line 305: While the algorithm is executed at Tx and Rx ends with specific steering angles, is there any possibility of synchronization error? If yes then please elaborate on it in the manuscript.
15. Page 11 Line 338: In comparison to previous research…. Please cite recent known cases to claim the improvement.
16. Result discussion is adequate – fair improvement is visible against complexity.
17. Figure 14: Fonts need improvement.
18. Conclusion: Limitations can be mentioned.
19. Overall, the manuscript is well organized.
Proofread is required.
Author Response
Comments are addressed in attached file.

Round 2
Reviewer 1 Report
The current version can be accepted.
Some parts are required to be revised.
Reviewer 2 Report
Comments are adequately addressed.